# Cortical Activity Linked to Clocking in Deaf Adults: fNIRS Insights with Static and Animated Stimuli Presentation

**DOI:** 10.3390/brainsci11020196

**Published:** 2021-02-05

**Authors:** Sébastien Laurent, Laurence Paire-Ficout, Jean-Michel Boucheix, Stéphane Argon, Antonio R. Hidalgo-Muñoz

**Affiliations:** 1Laboratoire Ergonomie et Sciences Cognitives pour les Transports (LESCOT), University Gustave Eiffel, IFSTTAR, F-69675 Lyon, France; laurence.paire-ficout@univ-eiffel.fr; 2Laboratoire d’Etude de l’Apprentissage et du Développement, Centre National de Recherche Scientifique (LEAD-CNRS UMR 5022), University of Bourgogne Franche-Comté, F-21065 Dijon, France; Jean-Michel.Boucheix@u-bourgogne.fr (J.-M.B.); stephane.argon@gmail.com (S.A.); 3Laboratoire Cognition, Langues, Langage, Ergonomie, Centre National de Recherche Scientifique (CLLE-CNRS UMR 5263), University of Toulouse, 31000 Toulouse, France; antonio.hidalgo-munoz@univ-tlse2.fr

**Keywords:** clocking, deafness, animation, fNIRS, motion prediction, temporal skill, time estimation

## Abstract

The question of the possible impact of deafness on temporal processing remains unanswered. Different findings, based on behavioral measures, show contradictory results. The goal of the present study is to analyze the brain activity underlying time estimation by using functional near infrared spectroscopy (fNIRS) techniques, which allow examination of the frontal, central and occipital cortical areas. A total of 37 participants (19 deaf) were recruited. The experimental task involved processing a road scene to determine whether the driver had time to safely execute a driving task, such as overtaking. The road scenes were presented in animated format, or in sequences of 3 static images showing the beginning, mid-point, and end of a situation. The latter presentation required a clocking mechanism to estimate the time between the samples to evaluate vehicle speed. The results show greater frontal region activity in deaf people, which suggests that more cognitive effort is needed to process these scenes. The central region, which is involved in clocking according to several studies, is particularly activated by the static presentation in deaf people during the estimation of time lapses. Exploration of the occipital region yielded no conclusive results. Our results on the frontal and central regions encourage further study of the neural basis of time processing and its links with auditory capacity.

## 1. Introduction

Current findings from research works dealing with deafness and temporal skills are often contradictory. The study of the temporal dimension is of interest because hearing is the sense which has the highest resolution for the estimation of time duration. It therefore appears to be essential for time processing, even when stimuli are presented via other modalities [1,2]. In addition, some experiments show that stimulus sequences are easier to recall when they are presented in the auditory modality [3,4]. 

These contradictory conclusions can be at least partially explained by the fact that “temporal skills” can encompass different notions, depending on the type of protocol used to examine time-related issues, as suggested by [5]. While some studies have evaluated the ability to replicate temporal sequences [6,7,8], others have focused on the ability to estimate time interval duration [5,9,10]. Results from the first category of works are extremely heterogeneous. However, there is more of a consensus in conclusions reached by studies in the second category, which have shown, for instance, that deaf adults experience difficulty in carrying out a time estimation in task based on visual stimuli [10]. Similar results were found in other studies using not only visual stimuli [9], but also the tactile modality [5]. Therefore, in order to elucidate the origins of the potential difficulties linked to time processing, it is essential to focus on specific mechanisms, such as clocking. Experimental tasks aimed at studying the ability to estimate time interval duration are usually either based directly on the estimation of a time period, or on time-to-contact assessments. Either way, these tasks are usually carried out in laboratory settings, with strict constraints for stimulus characterization. In addition, the variables used to draw conclusions are often linked to task performance, and to behavioural or subjective self-reported measures [11,12,13]. This could lead to biased results if confounding variables intervene in task execution. In some cases, therefore, the generalization of the results to daily life activities remains uncertain, and using them to aid deaf people becomes difficult. 

In order to overcome the above-mentioned limitations, in the present work we recorded brain activity during a decision-making task involving road scenes. Interest in the study of brain activity linked to daily situations, such as driving, where time estimation and motion prediction are needed, is currently on the increase. Earlier studies have, for instance, looked at electrical activity during overtaking, a situation in which accurate motion prediction is crucial to guarantee safety [14]. According to [12], motion prediction is the product of two cognitive operations: cognitive motion extrapolation (CME), and a clocking mechanism. In the case of driving, the first operation is based on a mental model of vehicle movement, and on its subsequent extrapolation in order to predict its evolution. The second operation, which we manipulated in the present study, consists of the advance estimation of time to contact. Both cognitive processes have temporal components, since the estimation of speed is essential to build a mental representation of dynamism for the CME, and a time judgment is needed for the clocking mechanism. 

In the present study we used the same protocol as in [15] to design two conditions, each with a qualitatively different clocking mechanism. These conditions were defined by the presentation format of the road situations, via either an animated video or a sequence of static images. According to the Attentional Theory of Cinematic Continuity (AToCC), the flow of information in animation makes construction of a CME easier than a motion sampled presentation, thanks to the attentional guidance of perceptual and cognitive processing provided by the dynamic presentation [7,16]. Several empirical studies have shown behavioral and eye-movement results consistent with the AToCC approach, for example, in the domain of multimedia learning [17,18], segmentation and comprehension of movies and films [19,20], visual narrative processing [21], and event prediction [22]. In contrast, the presentation of motion sampled by means of a sequence of static images, forces the subject to make more elaborate spatiotemporal inferences. In road situations, a time judgement and, therefore, a clocking mechanism is needed to infer vehicle speed when dynamic information is not provided continuously. Even though numerous behavioral studies have been carried out to infer these processes, physiological and brain measures can provide objective evidence of these difficulties or reorient the research towards other possible factors which may influence, for example, decision times or performance.

In the present work, we used modern functional near-infrared spectroscopy (fNIRS) to study brain activity linked to the clocking mechanism and to cognitive effort [23,24]. The fNIRS is a noninvasive technique that enables cortical activity to be inferred by measuring the light absorbed by the blood flow in the brain surface. It provides better temporal resolution than fMRI [25,26] (~250 ms), and is less sensitive to motion artifacts than the EEG [27]. Furthermore, fNIRS measurement is limited to the cortex and consequently no contribution of subcortical regions is expected in the signals, unlike EEG. The cortical area is the most relevant brain region for the study of the above-mentioned cognitive processes. fNIRS has the advantage of being portable and easy to set up, and is therefore a promising technique which will very probably be used in real life settings to monitor the cognitive state of users in the near future [28,29]. In addition, fNIRS is completely innocuous and has proven to be suitable in the field of brain-computer interface [30,31] through neurofeedback and also in the study of sensitive populations, such as newborns [32], deaf children [33] or Alzheimer Disease [34] and Parkinson’s [35] patients. Although this technique is usually employed in block-based analysis, an event related approach, such as the one we present here, is also suitable. Our framework allows us to determine whether cortical activity in different brain regions differs significantly in deaf people compared to people with normal hearing when the clocking mechanism is implemented [12] during the evaluation of road scenes.

To sum up, according to the findings from previous research works on the neural substrates of the clocking mechanism [36], and to the recent literature about attentional difficulties [37] and brain plasticity due to hearing loss or deafness [38], our hypotheses concerning the different brain regions studied (see Section 2.4. for details) are as follows: (1) evoked hemodynamic responses in the frontal region will be higher in deaf participants due to a greater cognitive effort when making spatiotemporal inferences [5,9]; (2) higher hemodynamic responses will be found in the central region for sequences of static images than for animations, because the clocking mechanism is required, and its underlying neural substrate is located in the pre-supplementary motor [36,39,40] and sensorimotor areas [41], both of which are located in the central brain region; (3) Because deaf people are attracted to spatial elements in tasks involving the estimation of time intervals [10] and because of their more highly-developed visual skills [42], we explored the occipital region to find out if there was any difference in the hemodynamic responses of deaf versus hearing participants, linked, for example, to deeper visual processing on the part of deaf participants, to alleviate the difficulties of time estimation. 

## 2. Materials and Methods

### 2.1. Participants

A total of 19 bilaterally deaf or severely hearing-impaired individuals (age: M = 28.1 years, SD = 8.8, 9 male) and 18 hearing individuals (age: M = 26.1 years, SD = 6.6, 10 male) participated in the study. Information about deafness onset, hearing aids and language of deaf individuals is presented in Table 1. All participants had normal or corrected to normal vision. None of them declared a history of psychological or neurological disorders. None of them had a cochlear implant. They were all taking lessons in traffic rules and preparing the Highway Code examination in order to obtain their driving license in an accredited driving school in France. All participants gave their informed consent for inclusion. The study was conducted in accordance with the Declaration of Helsinki, and the protocol was approved by the French Biomedical research ethics committee.

### 2.2. Task Involving Clocking

A decision-making task requiring spatiotemporal inferences was used. Participants were positioned in the driving seat and had to decide whether or not they had enough time to accomplish a driving action: overtaking, entering a roundabout, joining a highway and crossing an intersection, where other vehicles were involved. Participants had to respond “Yes I have time” or “No I don’t have time”, as quickly as possible, either during the presentation or at the end of the stimulus by using a keyboard with colored stickers (blue and yellow) placed on the two answer keys “S” and “L”. 

Road situations were presented for 8 s in two different formats: Animated, which consisted of an animated movie; and static image sequences, which consisted of three static pictures (each presented for 2.66 s) extracted from other animations (first, middle and last pictures). In order to make a decision when using the static format, participants needed to judge the length of time between images and estimate vehicle speed based on the evolution from the initial spatial cues. A schema of the stimulus presentation is depicted in Figure 1.

Four road situations requiring motion prediction were created: overtaking, entering a roundabout, joining a highway and crossing an intersection. These driving situations were modelled using 3D Studio software in accordance with French road norms. A judgment task, performed beforehand by expert drivers, was used to validate “yes” and “no” responses in complex situations. The experiment was designed in E.Prime 2.0 presented on a 23-inch screen (1920 × 768), in the same pseudo-randomized order for every participant. A total of 32 different stimuli were created for each format. The order was created in such a way that the same road situation, or the same format, was never presented consecutively. Before stimulus presentation, each trial started with a white fixation point in the middle of a black screen with a variable duration of {7, 7.5, 8, 10}. The variable duration of the fixation point can be considered as a jitter to avoid periodical physiological artefacts such as Mayer waves on the fNIRS signal (see Section 2.4 for fNIRS data processing). 

### 2.3. Experimental Setting

Participants carried out the experiment individually in a single session. Once the participant was seated comfortably, the fNIRS cap was put on his head and the light dimmed. A calibration was performed by means of NIRStar software version 15.3 to ensure good quality signals in terms of Signal to Noise Ratio and gain. Conductive gel was applied to the participant’s hair when necessary. Eye-tracking (Tobii TX300) was recorded simultaneously for further analysis. An over cap was used to prevent possible interference on the fNIRS signal from the eye-tracker. Instructions were then presented on a 23-inch screen, and explained orally to hearing participants, and in French Sign Language to deaf participants by a qualified sign language interpreter. The fNIRS recording lasted approximately 34 min on average, taking into account response times and baseline. 

### 2.4. fNIRS Data Processing

fNIRS data were recorded using a NIRScout device (NIRX Medical Technologies) with a sampling rate of 3.9 Hz. Two cap sizes were used, 54 and 58 cm, adapted to head size. This system has 40 optical sensors: 16 sources, emitting light at 760 and 850 nm, and 24 detectors. As a result, 52 channels were obtained by the combination of sources and detectors (Figure 2). 

Signal processing was carried out with MATLAB version R2019a (The MathWorks Inc., Natick, MA, USA, v. R2019a) with the Homer2 software package. After visual verification, channels without visible heartbeat oscillations (~1 Hz) or with large motion artefacts were considered as noisy channels and were discarded from analyses [43]. Pruning was then used to remove channels with low SNR [44]. Intensity values were converted into optical density. A band pass filter of 0.01–0.5 Hz was applied to attenuate noise from physiological changes (heart rate and breathing) occurring at high and low frequencies. A combination of Spline interpolation [45] and wavelet filtering [46] was made to correct motion artefacts, as proposed by [47]. The spline interpolation was used with *p* = 0.99 [45,48,49] while the wavelet had an IQR = 0.1. Optical density values were then converted into relative concentration changes of oxyhemoglobin: Δ(HbO); and de-oxyhemoglobin: Δ(HbR); using the modified Beer–Lambert law with a differential path length factor of 6 [50]. 

Event related responses were computed within 8 s segments, and a baseline of 1 s before stimulus onset was considered (see Figure 3 for an example). The approach of studying evoked hemodynamic responses can therefore be considered to be a rapid event-related study, in which the inter-stimuli intervals may be shorter than the elicited response [51]. As segments in the study were short, the mean signal value of Δ(HbO) was the most representative parameter [52]. 

Exploratory paired Student’s t-tests were performed to detect aberrant results and outliers. In order to avoid the influence of individual differences in placement, the channels were grouped into different clusters and the averaged signals were computed for further statistical analysis. Three regions of interest were defined: frontal, central and occipital, as depicted in Figure 2. Due to technical limitations it was not possible to cover the temporal region.

Figure 3 shows means Δ(HbO) and Δ(HbR) for the deaf group in the frontal region during the animated presentation. For the subsequent analysis, only Δ(HbR)was considered since it is supposed to be more sensitive to hemodynamic variations induced by cognitive changes than Δ(HbR), and is more frequently used [53]. Figure 3 shows that HbR value is indeed close to zero, which suggests the absence of any global blood flow artefacts due to head movements. The mean values of the Δ(HbR) during the 8 s windows were computed as the variables.

### 2.5. Statistical Analysis

Table 2 depicts channels discarded by region and participant, due to poor SNR, by the prune channel function in Homer2. Regions were not considered in future analyses when a minimum of 50% of their channels had been discarded. Δ(HbO) mean value distributions respected normality.

For whole-brain analyses, one-sample t-Student tests were computed channel by channel for every condition and for the two groups to verify whether mean values were significantly different to zero. Three repeated measure analyses of variance (ANOVA)-one for each brain region as depicted in Figure 2 were performed, where one within-subject factor: Format (two levels: Animation (ANI) and sequences of static images (STA), and one between-group variable (defined by deafness) were considered. Statistical analysis was complemented with planned contrasts according to our hypotheses between hearing and deaf groups for the STA condition. Statistical analyses were carried out using JASP v14.0 software.

## 3. Results

### 3.1. Whole Brain Analysis

Firstly, in order to determine which relevant scalp region was the most involved during the task, one-sample *t*-tests vs. 0 were carried out for every recording point (52 channels), for each group (hearing and deaf participants), and for every format condition (ANI and STA). Channels which were significantly different from zero are shown in Figure 4. More channels were activated in the STA condition for deaf people than in any other condition. Activation was seen mainly in channels from the frontal and central regions, but also appeared in the occipital area. In contrast, few channels were different from zero in the hearing group, where isolated channels were activated in parietal and frontal regions. 

To complement the previous results, a channel-wise analysis was performed to find the channels where significant differences existed between groups. As expected, numerous channels (numbers 5, 23–26, 28, 29, 34, 35, 36, 40, 41–44 and 48–50 represented in Figure 3) showed significant results for STA (*p* < 0.05). These preliminary results justify grouping the proposed ROI, as suggested in the literature [54]. 

### 3.2. Analysis Per Region

Figure 5 depicts the individual signals from the participants with the highest quality signals and the smallest number of discarded channels according to Table 2 (Participants D8 and H6), used as controls. Although the signal quality was high, the classical hemodynamic shape was not always found. However, a net typical shape of HbO in the central region can be observed in hearing participants. Full details of the main findings for each region are provided in the text below.

Although the significant results will be explained again in the text, Table 3 displays all *p*-values and related statistical parameters for each ANOVA.

#### 3.2.1. Frontal Region

After data processing, estimation of artefact contribution and channel pruning, 16 deaf and 12 hearing participants remained in the set for frontal region analysis. The signals from the other participants were discarded due to poor SNR. A main effect of GROUP was significant (F (1, 26) = 8.439; *p* = 0.007; η^2^ = 0.157). The planned contrast between hearing and deaf groups for the STA condition revealed that the responses of the deaf group to the STA format were significantly higher than hearing group (t (50, 27) = 2.945; *p* = 0.005). However, no significant differences were found for the ANI condition (t (50, 27) = 1.529; *p* = 0.133).

Figure 6 depicts the mean values of Δ(HbO) in the frontal region for each condition. Even when the mean value for the hearing group was lower than zero, the one-factor t-test suggested that this variable was not different from 0, contrary to the values for the STA condition in the deaf group, where the variable magnitude was significantly positive (*p* = 0.005).

In addition, Figure 7 illustrates the averaged signals obtained for static (STA) and animated (ANI) conditions in the deaf group in the frontal region. 

#### 3.2.2. Central Region

After data processing, 17 deaf and 13 hearing participants remained from the set for central region analysis. Only a main effect of FORMAT was significant (F (1, 28) = 4.281, *p* = 0.048, η^2^ = 0.029). 

Figure 8 depicts the mean values of Δ(HbO) in the central region. 

In addition, Figure 9 illustrates the averaged signals obtained for static (STA) and animated (ANI) conditions in both groups in the central region involved in the clocking mechanism. 

#### 3.2.3. Occipital Region

After data processing and estimation of artefact contribution, 9 deaf and 11 hearing participants remained from the set for occipital region analysis. It should be noted that, due to its particular anatomical characteristics, the occipital region is the most difficult area of the scalp from which to obtain good fNIRS signals. Neither the ANOVA nor the planned contrast for the STA condition (t (32.19) = 0.635; *p* = 0.530) revealed significant effects on the occipital region. The smaller sample size may have prevented us from obtaining significant results. 

Figure 10 depicts the mean values of Δ(HbO) in the occipital region.

Figure 11 illustrates the averaged signals obtained for static (STA) and animated (ANI) conditions in both groups in the occipital region.

## 4. Discussion 

The overall objective of the present study was twofold. Firstly, we aimed to verify whether the fNIRS technique could provide insight into the study of the clocking mechanism involved in motion prediction. In order to do so, we used animation, along with another format consisting of sequential images, which required the addition of a clocking mechanism. Our second aim was to analyse the difference in the hemodynamic responses evoked by these stimuli in deaf people, and people with normal hearing. The results suggest that relevant conclusions can be made on our hypotheses about the brain regions studied.

Our first hypothesis concerned the frontal region. Based on previous research showing that deaf people experience difficulty when estimating time intervals, we expected to observe greater activation when they were asked to make a spatio-temporal inference [5,9]. We found that evoked response in the frontal region was higher in deaf participants than in their hearing counterparts only in the condition where a sequence of images was presented and an additional time estimation between images was required, whereas no difference was evidenced for the animations. As Figure 7 shows, the classical hemodynamic signal from fNIRS was obtained in the deaf group, while a more complex response was obtained in the static condition. This hemodynamic response is usually associated with the trigger of determined cognitive processes in determined regions. In the case of the frontal and prefrontal regions, it is linked to a wide range of cognitive mechanisms including executive functions. It is, therefore, a reliable measure of cognitive effort, as suggested in other studies [23,24]. However, it is more difficult to reach any conclusions from the mean average of the hearing group, shown in Figure 7, since an unstable signal seems to appear. The mean values are not statistically different from zero for this group. This suggests that the task probably required less cognitive effort and was performed quickly, which in turn implies that participants disengaged rapidly after making their decision.

The reasons for this difference in effort in deaf candidates may be varied, and deeper brain analysis is required to determine the specific processes. One possible explanation may be related to the difficulties in time processing, and in subsequent spatiotemporal inference and motion prediction, shown in a number of studies [5,9]. Nevertheless, caution is required, since many cognitive processes take place in the frontal region, and in the case of deaf people, recalibration processes and specific plasticity characteristics are involved. This fact motivates the study of more specific regions which are thought to be more closely linked to time cue processing, such as preSMA, discussed below.

Our hypothesis on the central region, which has been particularly linked to the clocking mechanism [11], is in agreement with the present results, which showed that activation occurred during the estimation of time. Interestingly, two main conclusions can be drawn from the grand average shown in Figure 9. Firstly, in the case of the static format, two consecutive peaks were elicited. This phenomenon was also found in the individual signal shown in Figure 5 for deaf participants. Although the response delays here seem very short in comparison to the usual hemodynamic response latencies, we cannot dismiss the possible contribution of the two clocking mechanisms in the static condition. Unlike the animated condition, in this format, dynamism is not transmitted directly. In its absence, participants tried to mentally infer the speed of the vehicles from the spatial elements with a first clocking mechanism. Alternative interpretations could also be considered. Processes which disrupt attention, as described in the AToCC model, could be linked to the reactivation of attentional resources when a second picture is presented. However, if the contributions to these peaks were linked exclusively to automatic processes, no difference in the signal shape would be expected compared to the hearing group. Indeed, the average of all participants showed a similar shape, with a lower amplitude in the hearing group. This could be linked to lower cognitive effort or less attentional disengagement. In order to nuance these statements and ensure that they are not only speculative, it is crucial to pursue the research and include other, similar tasks.

The peak found in the central region in the animated condition could be linked to the clocking mechanism required to predict vehicle movement, if we focus on our hypotheses and on the justification of our work. The continuity of information provided by animation would therefore require the intervention of only one clocking mechanism. This explanation is also in agreement with the AToCC theory, as the introduction of intervals can trigger ruptures that disrupt continuity [16,17]. No significant evoked responses were found in people with normal hearing. Their clocking mechanism is probably more developed and the effectiveness of the process does not allow evoked responses to be obtained. However, we consider that it is important to note the surprising typical hemodynamic response in the hearing participant with the highest quality signals (in terms of gain level, SNR and artifact contribution) depicted in Figure 5. This signal, which was reproduced in the other channels composing this region for this participant, hints at a specific common processing in both conditions, which could arguably be linked to spatio-temporal inferences. Either way, the activation of this region may have been impacted by the motor preparation prior to response. However, this influence would be comparable in both presentation formats, and would consequently have been cancelled out once all the trials had been averaged. Further analyses, where more spatial and temporal resolution are available, are needed to confirm this hypothesis.

The exploration of the occipital region did not provide clear results. Despite the fact that several studies have shown more highly developed visual capacities in deaf individuals [55] and an attraction to spatial elements in tasks involving time interval estimation [10], the fNIRS technique did not evidence this using the present protocol. It could be argued that the present measure of global activity in the occipital region would not really show the preferential processing by deaf participants of the spatial features in visual scenes. It is possible that a compensation mechanism acts primarily within the temporal region, as suggested by numerous studies [56,57,58]. Moreover, as mentioned above, the occipital region is the scalp area from which it is most difficult to obtain good signals, and another functional technique would probably be more suitable than fNIRS.

Our work presents a number of limitations. The inherent characteristics of each road situation in each environment could lead to qualitative differences when making the CME, mainly in the case of sequences of images. This can be particularly observed in the case of joining a highway, which involves the visualization of a rear-view mirror and lateral vision. However, this issue is not supposed to interfere with the clocking mechanism, and makes the study more ecological and complete for a wider range of driving situations. It would have been interesting to study activity in the temporal region, especially in relation to brain plasticity, and because the auditory cortex is situated in this region. However, the area is characterized by high levels of blood flow artefacts and there are technical limitations on the number of sources and detectors which can be used (16 sources and 24 detectors). There is also no evidence in the literature of its involvement in the clocking mechanism. For these reasons, we chose not to include it in the present study. A larger sample size might have provided more generalizable results and may have allowed the study of specific deaf characteristics. In future works, a subgroup of the deaf population could be studied with a view to designing specific aids. Conducting studies using fNIRS in ecological settings could also enable these processes to be monitored in different contexts.

## 5. Conclusions

To sum up, these preliminary results encourage further analysis of the neural basis of the clocking mechanism in the deaf population, since they show higher hemodynamic responses in both frontal and central regions when time estimation is required to determine vehicle speed. The experimental protocol based on animations and image presentation has proved to be suitable for this goal. These promising results suggest that time estimation could be analyzed by using the fNIRS technique in central and frontal regions to complement performance measures using different approaches. In addition, the example used in this protocol, which was extracted from pictures from the Highway Code test, highlights the potential difficulties that should be considered when designing material for learning and the evaluation of skills in deaf people.

## Figures and Tables

**Figure 1 brainsci-11-00196-f001:**
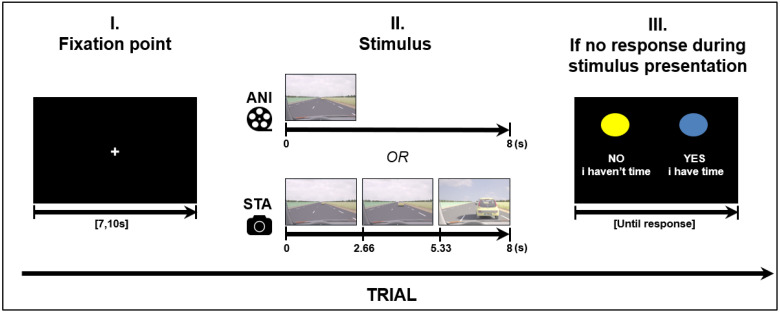
Structure of a trial (road scene presentation), ANI: Animated format; STA: sequence of static images.

**Figure 2 brainsci-11-00196-f002:**
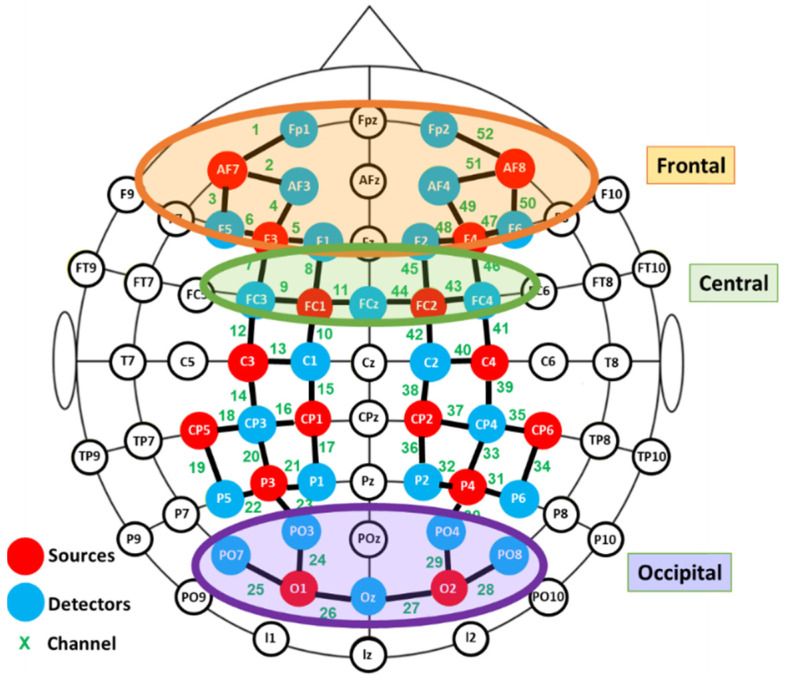
Functional near infrared spectroscopy (fNIRS) montage. Signal acquisition was made by the colored optodes.

**Figure 3 brainsci-11-00196-f003:**
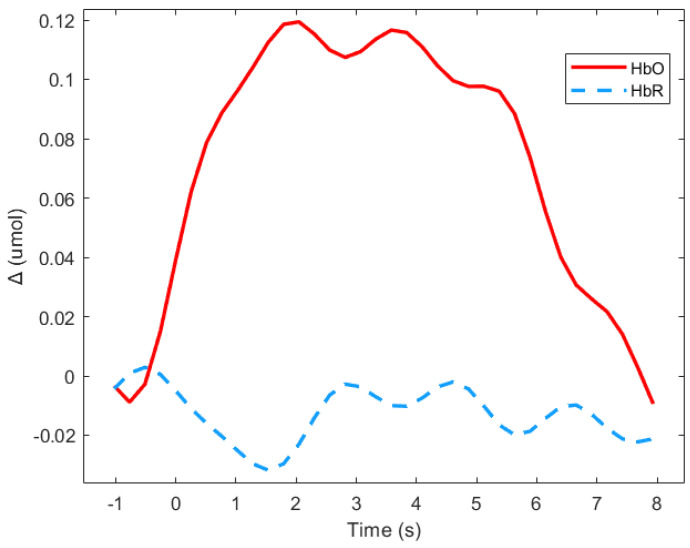
Average of the hemodynamic evoked response in the deaf group during animated road scene processing in the frontal region.

**Figure 4 brainsci-11-00196-f004:**
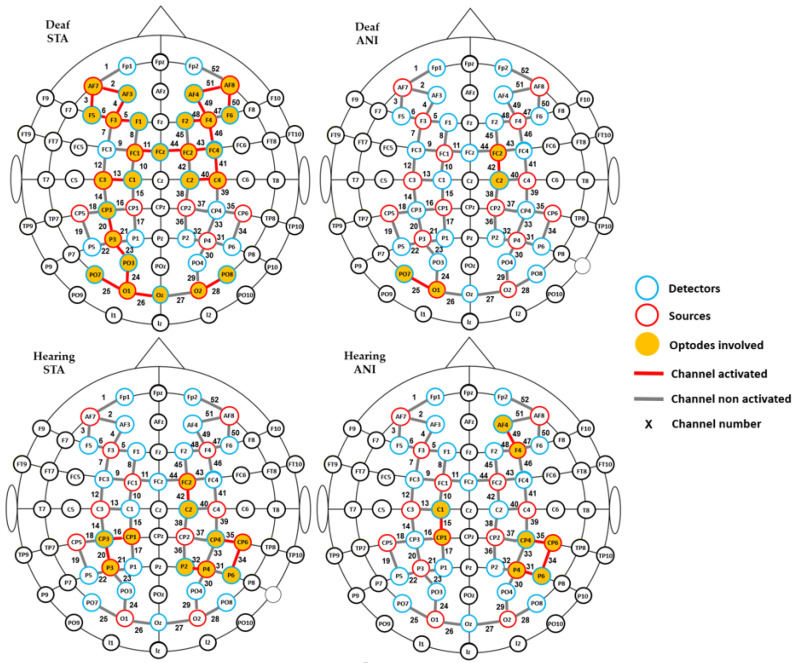
Channels with significantly positive activation for each group (deaf and hearing) and format (STA: sequence of static images; ANI: Animated format).

**Figure 5 brainsci-11-00196-f005:**
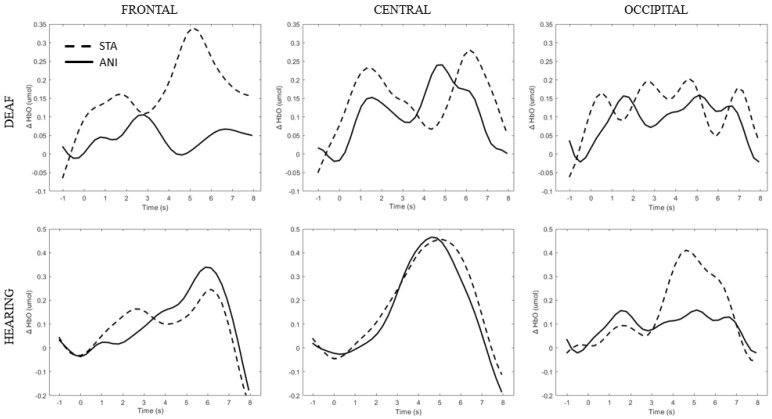
Average of the hemodynamic evoked response for one deaf (D8) and one hearing individual (S6) for each region (frontal, central and occipital) and each format (STA: sequence of static images; ANI: Animated format.

**Figure 6 brainsci-11-00196-f006:**
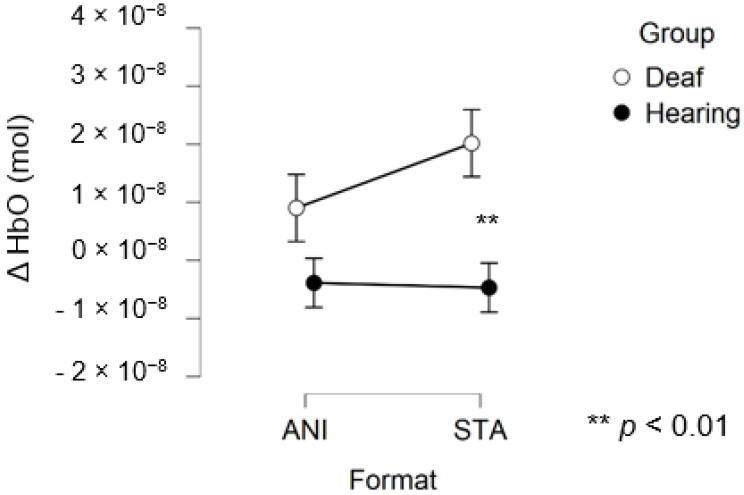
Mean amplitude values of the hemodynamic evoked responses in the frontal region for animated (ANI) and static image sequence (STA) formats for both groups.

**Figure 7 brainsci-11-00196-f007:**
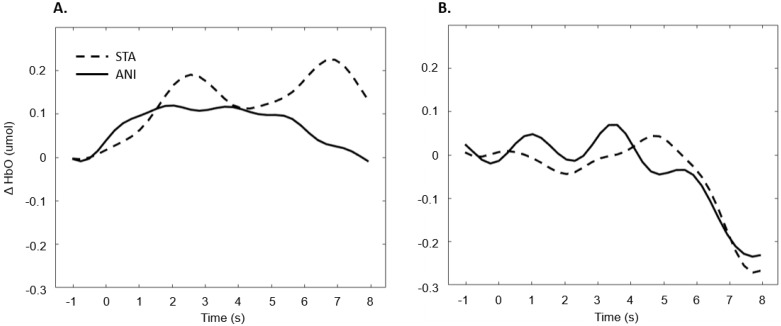
Average of the hemodynamic evoked response in the deaf group (**A**) and hearing group (**B**) for the two conditions in the frontal region.

**Figure 8 brainsci-11-00196-f008:**
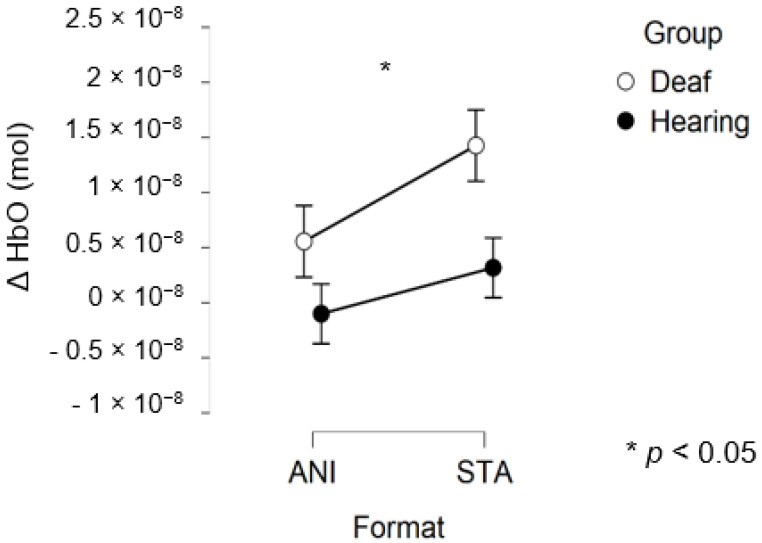
Mean amplitude values of the hemodynamic evoked responses in the central region for animated (ANI) and static image sequence (STA) formats for both groups.

**Figure 9 brainsci-11-00196-f009:**
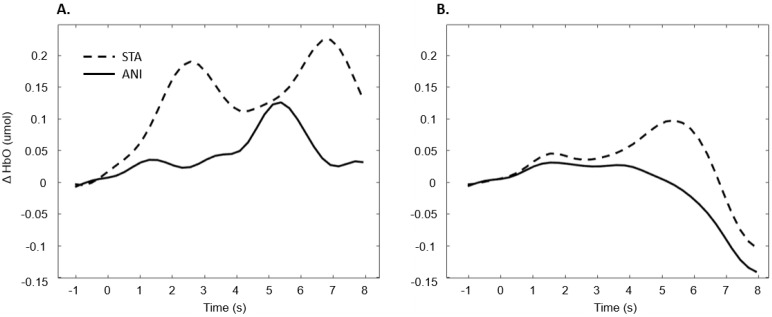
Average of the hemodynamic evoked response in the deaf group (**A**) and hearing group (**B**) for the two conditions in the central region, which is involved in clocking mechanism.

**Figure 10 brainsci-11-00196-f010:**
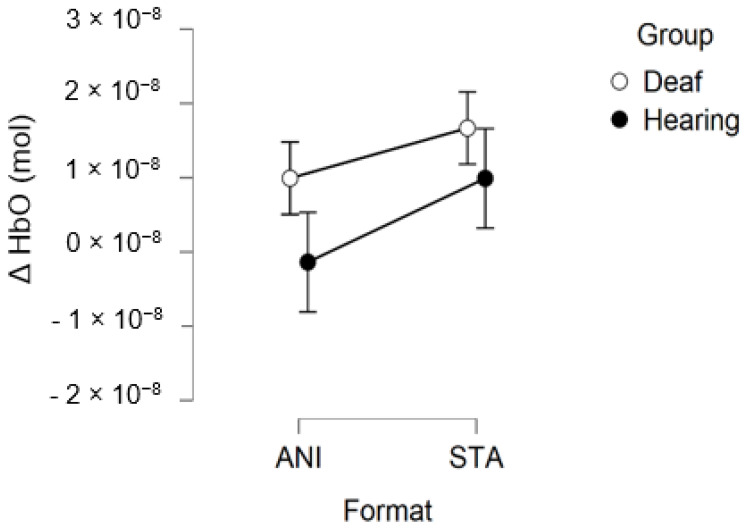
Mean amplitude values of the hemodynamic evoked responses in the occipital region for animated (ANI) and static image sequence (STA) formats for both groups.

**Figure 11 brainsci-11-00196-f011:**
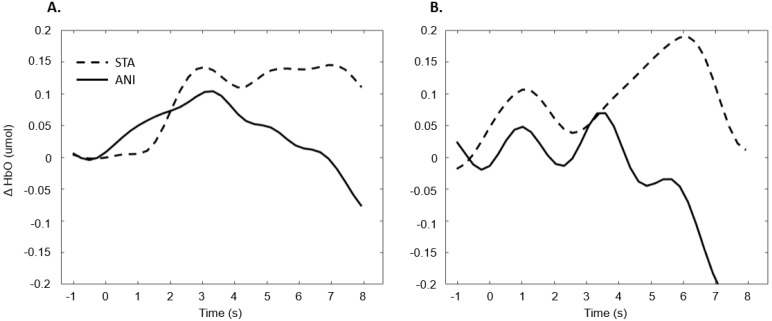
Average of the hemodynamic evoked response in the deaf group (**A**) and hearing group (**B**) for the two conditions in the occipital region.

**Table 1 brainsci-11-00196-t001:** Demographic information about deaf participants.

Participant	Deafness Onset	Hearing Aid Used	Language Used
S1	From birth	Prosthesis currently used	Oral and French sign language
S2	Before 2 years old	Prosthesis currently used	Oral and French sign language
S3	From birth	Prosthesis currently used	Oral and French sign language
S4	From birth	Prosthesis currently used	Oral and French sign language
S5	From birth	Prosthesis currently used	Oral and French sign language
S6	From birth	Prosthesis currently used	Oral and French sign language
S7	From birth	Cochlear implant used in the past	Oral and French sign language
S8	From birth	Prosthesis currently used	Oral and French sign language
S9	From birth	Prosthesis currently used	French sign language
S10	After 2 years old	Prosthesis currently used	Oral and French sign language
S11	From birth	Prosthesis currently used	Oral and French sign language
S12	After 2 years old	Prosthesis currently used	Oral and French sign language
S13	Before 2 years old	Prosthesis used in the past	French sign language
S14	Before 2 years old	Prosthesis currently used	Oral and French sign language
S15	Before 2 years old	Prosthesis currently used	Oral and French sign language
S16	From birth	Prosthesis used in the past	French sign language
S17	From birth	Prosthesis used in the past	French sign language
S18	From birth	Prosthesis currently used	Oral and French sign language
S19	From birth	Prosthesis used in the past	French sign language

**Table 2 brainsci-11-00196-t002:** Channels discarded by participant and region (see Figure 2 for channel identification). Channels are in bold when they belong to discarded regions (more than 50 % of channels were withdrawn due to artefacts).

Participant	Frontal (1–6; 47–52)	Occipital (24–29)	PAreSMA (9–13; 40–44)	Discarded Region (s)
D1	49	24–28	44	Occipital
D2	None	**25; 27; 28; 29**	None	Occipital
D3	1	None	None	
D4	4; 5; 47; 48	**27–29**	None	Occipital
D5	None	None	44	
D6	3; 6; 52	None	11	
D7	1; 4; 5; 47; 52	None	12; 44	
D8	6	None	None	
D9	None	**24; 28; 29**	42	Occipital
D10	**1–47; 49; 52**	**24; 25; 27; 28**	41	Frontal and occipital
D11	47; 50; 52	None	None	
D12	**3–5; 47–52**	**24–29**	All	Global
D13	1–3	**27–29**	None	Occipital
D14	1–3	None	None	
D15	None	**27–29**	None	Occipital
D16	**1–5; 47–50**	**24; 26; 27**	**9–12; 41–44**	Global
D17	1; 4	**24–29**	12; 40	Occipital
D18	None	None	None	
D19	None	29	None	
H1	None	All	None	Occipital
H2	**1–6; 47; 49; 50; 52**	25; 26	**9; 10; 12; 13; 41–44**	Frontal and central
H3	49	None	None	
H4	47	28	None	
H5	**1–4; 6; 47–52**	26–29	**9–13; 4–44**	Global
H6	None	None	None	
H7	None	**24; 26; 27**	10;13;42	Occipital
H8	None	None	None	
H9	**2–4; 6; 47; 49; 51; 52**	27	10;12;13;41	Frontal
H10	**1–6; 47–52**	**All**	**9–13; 40–44**	Global
H11	None	26	None	
H12	1;47;52	None	None	
H13	None	**26–29**	None	Occipital
H14	5;52	**24; 26; 28; 29**	11;43;44	Occipital
H15	**2; 5; 6; 47; 49; 52**	**25–29**	**9–12; 41–44**	Global
H16	47;52	None	None	
H17	47;52	None	11;40;42	
H18	**2; 3; 6; 47; 50; 52**	27; 28	**12; 13; 40; 41; 43; 44**	Frontal and central

**Table 3 brainsci-11-00196-t003:** *p*-values and related statistical parameters for each ANOVA.

	Main Effect		Interaction
	FORMAT	GROUP	FORMAT × GROUP
Frontal	F (1, 26) = 0.917, *p* = 0.347, η^2^ = 0.012	F (1, 26) = 8.439, *p* = 0.007, η^2^ = 0.157	F (1, 26) = 1.231, *p* = 0.277, η^2^ = 0.016
Central	F (1, 28) = 4.281, *p* = 0.048, η^2^ = 0.029	F (1, 28) = 2.148 *p* = 0.154, η^2^ = 0.055	F (1, 28) = 0.528, *p* = 0.473, η^2^ = 0.004
Occipital	F (1, 18) = 2.180, *p* = 0.157, η^2^ = 0.037	F (1, 18) = 1.069, *p* = 0.315, η^2^ = 0.037	F (1, 18) = 0.137, *p* = 0.715, η^2^ = 0.002

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
