# Peer review of "Cortical Activity Linked to Clocking in Deaf Adults: fNIRS Insights with Static and Animated Stimuli Presentation"

_brainsci, 2021, doi:10.3390/brainsci11020196_

Round 1

Reviewer 1 Report

In this paper authors explore brain hemodynamic responses by comparing a group of deaf and normal hearing participants, by using a novel method of infrared spectroscopy (fNIRS) after presentation of a road scene in animated format or by using a sequence of static images. The experimental paradigm and fNIRS methodology are fine. However results, at least in present format, do no supports any relevant scientific conclusion.

Weakest points and suggestions

To divide the brain in three coronal pieces is a consequence of the methodological approach, however it may afford wrong results. Analysing all recordings points together (brain as whole), if are statistically significant, could give a more robust results. Information about the temporal area, not analysed in this paper, is very relevant for comparison between deaf and normal hearing participants. This is because, auditory cortex is in fact the area, which drives differences among groups.

Sensory recalibration in deaf people act as a result of intermodal interactions, which indeed affect cortical visual processing, after maladaptive plastic reactions which allows to preserve cortical visual processing in absence of auditory cortex intrinsic horizontal regulation. Such altered cortical visual output indeed drives the hemodynamic responses in the behavioural experimental paradigm presented in this manuscript. In other words, deafness indirectly induces changes in visual cortical responses delaying recalibration from and to associative cortices. Such changes, and others as alterations in sensory motor gates and pre motor processing, underlie the cortical hemodynamic responses when a sequence of static images is presented.   So, because cross modal plasticity after deafness is a dynamic process along time, differences of sensory recalibration can be expected depending on: The duration of sensory loss, if deafness is unilateral or bilateral and if patients have been use or not sensory aids or cochlear implants. Because of this, a more exhaustive description of patients (audiogram, evoked potential, time duration of deafness, etc.) needs to be added to assess if the sample of participants is homogenous in the moment when the experiments were conducted.

A detail description of the cohort of deaf participants and a statistical comparison between results from rostral, middle and caudal recordings could be necessary, despite authors will try to analyse in the future the brain as whole.

-As a control, a few individual graphs of hemodynamic responses in deaf and normal hearing patients at the three brain regions for analysis must be provided.

-Authors must explain in more details why a high number of participants were discarded. Poor SNR in this group of participants is due to lower activation of participants or by technical problems of the set up?

Basal activation of normal hearing participants (5 sec before behavioural paradigm stimulation) must be used as a control for the statistical analysis by using multiparametric ANOVA T test.

-Graph in Fig4, 5 and 7 needs a little more explanations in the legends and in the text.

-Basal level of light absorption is around 0.0e-00 1.5e-08 in the middle brain region (Fig 5) and 0.0e -08 to 2e-08 in rostral and caudal recordings. These differences must be discussed and statistically analysed by comparing between cortical regions and between groups.

In sum, I suggest before publication, to modify the way for data analysis, a separate specific and extensive discussion of the results and a new more clearer conclusion section.

Reviewer 2 Report

Thank for your submitting this interesting study. The manuscript presents an interesting study on temporal processing abilities in deaf people which might have potential implications on the design of training aids for such people. Also the use of fNIRS might allow us to "directly" without bias measure the underlying mechanisms that support our ability to process spatio-temporal information in daily life. These results would add to our knowledge about this population which might help us better serve them. However, there are some major concerns I would like you to address before we can reconsider this manuscript for publication.

The hypothesis are not substantiated in the introduction. The introduction needs to explain in more detail why we need this particular study and how the study helps us answer the experimental questions. For example, it is not clear why there are three regions. A fronto-central giant region can be sufficient to see both cognitive processing/clocking mechanisms. Given the spatial resolution of fNIRS and the lack of data from temporal and more central/lateral areas that represent sensorimotor/premotor activations it is hard to understand why you chose to have three regions. It might just be my misunderstanding the text but the hypothesis about the occipital region seems to be vague and unsubstantiated. A clearer explanation might help me understand that better.

No information about the deaf participants is provided. It is important to consider details like onset/duration and level of hearing loss while assessing this data. it is not clear how representative this sample is of the general population of deaf people. These factors are known to greatly impact the plasticity and neural mechanisms that are being probed in this experiment.

Though you mention that the situations used in the testing were validated by expert users, there is no mention of how the responses from the test participants was validated. How do we know that the subjects were on task? Does that matter? Also, a minor detail about how you ensured that the participants maintained attention on the screen for the whole 8 seconds might be important for the reader to consider. For example, it is absolutely necessary that the subjects saw the second image. If they missed it, how does that impact the change in HbO.

Your conclusions about having two clocking mechanisms might not be the only possible explanation for your results from the central region. You might need to provide more support for this conclusion and possibly weigh this against some alternatives.

I would recommend adding some more details to the introduction to help situate the study and about fNIRS to help readers quickly consider the results. For example, the disagreements between behavioral data can be explained by differences in modalities of stimuli, test parameters, populations etc. Also, the studies you cite analyze different possible temporal mechanisms. So, it is essential that you provide bring together these details and precisely define what temporal mechanism you are interested in. Currently the introduction does not help me quickly understand the background and need for this study.

You need to separate the results from the discussion. The inclusion of the discussion section might allow you to comprehensively discuss the implications of your results and help the reader contextualise them. See https://www.mdpi.com/journal/brainsci/instructions#manuscript for required sections.

Please see my detailed comments below for details about some of the above comments. I look forward to reading an updated version of your manuscript.

Detailed Comments

Abstract

Seems to suggest that only the static scene requires the clocking mechanism.

P1. L9. temporal processing cues

P1. L11. "is to analyse the brain activity underlying time estimation"

P1. L16. Consider adding more detail about static presentation for example like, “using a sequence of 3 static images that presented the beginning, mid-point, and end of a situation…”

P1. L17-18. Consider revising to first present the result and your conclusions à for example, “the results show higher frontal region activity in deaf people which suggests greater cognitive effort….”

P1. L19. “involved”

P1. L18-20. Consider splitting this sentence. It is hard to understand since there are too many concepts presented in a long sentence.

Consider adding results for the occipital region even though you did not find any results. You can add a mention why you think your hypothesis did not hold.

Introduction

P1. L 28. Consider revising “desirable to deep into ….”

This opening statement is unsupported and would be better presented after the said evidence has been presented in the manuscript.

P1. L 35. “process (the) time dimension” It is unclear what this phrase means since the different studies cited here study different aspects of temporal processing. I think a more nuanced discussion of what kinds of temporal processing tasks is required rather than a “time dimension” especially since this paper investigates a clocking mechanism completely different from for example the fine structure processing deficits studied in the Moore et al. (2006) study.

P1. L 33-36. There are many differences between modality of stimuli studied, populations tested, methods used, metrics used etc. in your source. For example, in [8,9] they only study congenitally deaf adults which might have important implications for the origins of the deficit.

I would recommend a more comprehensive discussion of the literature to support why you think it is important to study temporal deficits in deaf people and how these studies actually point towards an underlying temporal processing deficit in this population which merits consideration.

P2. L45-47. The biggest difference your study offers is the use of brain activity (in deaf individuals). It is not clear why using brain activity to study daily activity using road scenes is necessary. If you can study behavior in these situations why use physiological responses?

P2. L 48/L55. You use two different ways to cite past work. This is inconsistent.

P2. L52. Consider replacing “in” with “of”.

P2. L55. “as Laurent….”

P2. L59. “thanks …. ”

P2. L 57-59. Are there any peer-reviewed references for the theory of Attentional Theory of Cinematic Continuity (AToCC)? You 2020 paper studies this in hearing impaired participants. Can you please provide references to other peer-reviewed work that shows that this idea has been validated in normally-hearing adults in this driving (or at least other) contexts? It is important to establish any support for the baseline data.

P2. L60. “elaborate”

P2. L63. Consider leading with the idea of easy portable testing instead of novelty. It might ground the study much better than just trying to use fNIRS to study a paradigm.

P2. L63-. Consider providing some references for the fNIRS technique and some context for what physiological responses are used and how they can be used in studying the underlying processes that support behavior. Even a brief presentation of this information and references might really help readers who are unfamiliar with this technique or clinicians who are not aware of the latest methods.

Also, your framing seems to suggest that fNIRS is used solely for the spatial localization of the clocking mechanisms. Is this accurate, can you describe how the specificity of the method can be used in this study?

P2. L67-69. Can you please list what areas of the brain are of interest to understand this clocking mechanism? This might help orient the reader’s expectations of possible hypotheses.

P2. L 73-76. Can you please provide a physiological/imaging study to support this hypothesis? Reference 24 is based on behavior. Are you suggesting some sort of sensory compensation would mean that deaf people would have greater visual activation?

It is not clear why you expect a difference in Occipital activation in the deaf group compared to the hearing group?

Materials and Methods

Participants

Did you collect any data on the etiology of the hearing loss from the deaf participants? Was any audiological evaluation conducted to quantify the level of hearing loss/time of deafness? Were they all congenitally deaf? Did they use hearing aids? Use of early interventions like hearing aids is known to affect brain plasticity and might impact your results. These details might be vital in the interpretation of your results. The possible variability in the data from these details might be important. These are also vital for replication of your results.

P2. L89-91. What does each button represent?

P3. L104. What was varied between the different stimuli? Were there only 4 road situations used?

P3. L116. Why did the testing take 34 minutes? What was the time between stimuli? Were the subjects allowed breaks for rest? These details are vital for replication.

P4. L138. “discriminate aberrant results”

P4. L139. “would not be possible?” Introducing the fNIRS and its possible limitations in the introduction might front-load these details and avoid distractions here.

P4. L142-143. Given your hypothesis about the central region-of-interest (ROI), do you think your selection of electrodes for this ROI and lack of temporal responses might limit your ability to measure potential clocking mechanisms?

P4. 143. Can you please detail what technical limitations?

P4. L144. Is this from an individual or a mean from the group?

Figure 2. Were only the filled-in sources and detectors used in the study? Why not present response during the static and animated sequences rather than the HbR which you do not use in your analysis? Also consider adding the hearing group to this figure. This would make it similar to Figure 6.

Results

Did the data meet normality requirements? Were there any outliers and were they dropped from the analysis? Please consider including a table with all results in them to allow the reader to examine all tests performed and results for all three ANOVAs. Also mention what the planned contrasts were. Without this it is hard to understand what you tested and if any corrections were performed to accommodate multiple comparisons? If you did adjust the p-values please mention what method was used wherever applicable.

Frontal Region

Was the effect of format significant? It looks like the two groups are different in the ANI condition, so I am curious as to what this result is?

P5. L161. Please place the “2” for the squared operation in superscript.

P5. L 163-166. Please mention both t and p values for both results. Did you test the differences between the groups in the ANI condition?

P6. L170. What does “important” mean?

P6. L171. As stated earlier it is important to see the time course of the evoked response in the animated condition.

Central Region

P6. L178-179. This statement seems to trivialize this choice and the whole central region hypothesis. Why not use just one large region that includes both frontal and central regions? This could help you set up process-based hypotheses instead of region-based hypotheses? This way you could have fewer/clearer things to test.

P6. L181-182. Please clarify/correct this sentence? How did you test deaf group vs animated condition?

Please mention t and p values for all t tests.

Please consider including the time course of the responses from the hearing group to allow the reader to see any dynamics. Also, even though the normally hearing listeners might not expend the extra resources in processing the event, they still have similar mechanisms and it would be useful to see the dynamics of that to get an idea of what the baseline is.

P7. L194-195. Could one argue that there are twoish (multiple) peaks in the ANI condition too?

P7. L195. Can you qualify “short” here?

P7. L 195-197. For the first mechanism from the participant estimating the time between the first and second image presentation? What purpose does this estimation serve? You only mention one possible clock mechanism in the introduction which helps us build the CME. Why would we need two clocking mechanisms?

Are there other possible explanations? Can the same timing mechanism be triggered twice?

This explanation might also line up with the AToCC, the introduction of change might trigger mechanisms that break continuity. The timing of the first peak does seem to line up with the presentation of the second image in the static condition, so is it possible it is just an effect of stimulus novelty?

Also, as stated earlier there might be multiple peaks in the ANI condition. Also, the response in the ANI condition seems to drop off much more and stabilize over the last portion of the 8 second window?

P7. L201-204. Would the inability to see the responses in the hearing group limit the ability of the fNIRS technique and/or impact our ability to interpret the results of this study? Please explain.

P7. L205-207. Should this be moved to line 203 before the last sentence? It seems out of place here.

Occipital Region.

In spite of the difficulties in obtaining the results from this region you obtained responses in 20 out of the 37 subjects.

Is the response in the normally-hearing group in the STA condition significantly greater than 0?

Can you please plot the time course of the response for both groups similar to Figure 6 for this condition?

It is not clear how more developed visual capacities would affect the response in the deaf group? All the participants irrespective of group association had normal hearing and no measure of visual acuity was obtained, so it is really hard for me to understand what would change here.

P8. L219-221. What is being compensated? There was no auditory component to the stimuli. What processing are you referring to?

Supplementary materials – is this incomplete? Please update appropriately. Also, I was only able to see videos of the animated condition in the supplementary materials provided to me. Do you need to upload more of these videos?

References

Some of the references are possibly in French. You might have to translate them to English.

Reviewer 3 Report

We consider the study carried out in this work interesting and well developed since the experimental setup is very accurate in detail.

The use of the fNIRS technique to study neuronal activity is a method increasingly used because of its high manageability, absence of side effects and the low cost that allows an innovative approach. However, it is not yet a standardized method, which negatively affects both the reproducibility of the proposed studies and the interpretation of the data. Taking into account the results obtained, particularly with regard to those about the frontal region, where a significant difference between the normal hearing group and the deaf one was found, the scientific  substrate needs greater specificity. Since frontal and prefrontal areas are involved in many cognitive processes, it is premature to state that difficulties in time processing could be involved in a univocal way. In conclusion, we consider a good attempt of investigation and research about the possibilities of the use of the fNIRS technique to derive from the cerebral hemodynamics information concerning the specific neurological functioning.  We recommend improving the work by considering expanding the sample and/or to investigate the possible scientific correlations between the data obtained with fNIRS and the probable time processing mechanisms involved.   Check some typo (line 19: involved)

Round 2

Reviewer 1 Report

Data analysis has been improved, in particular after including a list of hearing status of participants and new resulys about channels activation. Is evident that the authors have been made a strong effort to improve the manuscript. This is a methodological approach with many technical limitations  for reaching roboust scientific conclusions, however the novelty of the method make the paper interesting for the scientific community. I suggest to revise carefully again the manuscript to check for small mistakes as Table X instead Table 1 in line 206.

Reviewer 2 Report

Thank you for the comprehensive overhaul of the paper in response to comments from the reviewers. The paper reads much better now and is organized better. Several important details (like information about the deaf group) have also been clarified. I would still agree that the data presented here might be a novel application of fNIRS to a new problem and thus potentially interesting to researchers interested in using physiological responses to study this problem. However, I would like you to clarify some major questions that still remain before this manuscript can be published.

The ROI-based hypothesis still seems under substantiated and a result of the methodological limitations rather than a priori assumption based on current knowledge. The support for these regions is limited to the few sentences on Page 2. Also using a one sample t-test performed on the group mean responses comparing to zero does not support the need for these regions. Also, a lot of the electrodes listed on Page 7 L 198-199 are between the selected central region and occipital regions. How did you decide to limit the central region based on these electrodes? You also cite a "best practices" article to support your choice of electrode grouping. Can you clarify that?

For including the occipital region, you cite that deaf adults prefer using available visuo-spatial cues for processing temporal information and have more developed visual skill. These do not necessarily need differences in occipital lobe activation (for example plasticity in auditory areas). Your citation for the superior visual skills in deaf adults only shows improved peripheral processing rather than overall superiority in visual processing.

Is there a baseline level of response expected in each region and how does that affect how you have analyzed the data?

You have much fewer hearing subjects in the analysis of the central region. So, could this affect your ability to capture the proper response in this group?

The new additions seem especially rushed and not proofed as well. The whole paper needs careful editing and proof-reading to not only eliminate mistakes and missed changes (for example, the word important on P11. L274) but also avoid some of the awkward language. The text is really hard to read in its current state and I had to re-read several portions to understand them. The manuscript could greatly benefit from a thorough review by native/expert speakers.

Does greater frontal lobe activation only mean greater cognitive effort? It could also mean they are processing more information or preparing to process changes in visual information.

Minor Points

When you use “two clocking mechanisms” are you positing that the participants estimate time twice, once based on first and second image and then again based on the second and final image?

Some of the examples of static images (in the supplement) seem like they might not actually provide only information about motion of objects in the scene. For example, in the highway situation, the second image is of the side of the road (so just a blurry picture of the rails by the side of the road). How does this actually convey movement? Could this perhaps muddy the responses? Why use different scenes when you have not even established if this method can be applied to this problem?

Even though the whole-brain analyses were done post-hoc, it would help if the description of all statistical methods are presented in the analysis section.

 It is not clear what Figure 5 adds to the paper without similar information about the variability in responses. Could this information be moved to a supplement and just described in the discussion. Can you please add variability to the hemodynamic responses?

The heading titled “Supplementary section” Still contains “The following will be available online.” Is that incomplete?

The list of references contains a list of special characters like “«”. The word “No.” to indicate journal number is displayed correctly. Some of the months are not in English for example “mars 2010”.
